# Unlocking New Avenues in Breast Cancer Treatment: The Synergy of Kinase Inhibitors and Immunotherapy

**DOI:** 10.3390/cancers15235499

**Published:** 2023-11-21

**Authors:** María José Bravo, Antonio Manuel Burgos-Molina, Marilina García-Aranda, Maximino Redondo, Teresa Téllez

**Affiliations:** 1Surgery, Biochemistry and Immunology Department, School of Medicine, University of Malaga, 29010 Malaga, Spain; mjbravo@uma.es (M.J.B.); aburgos@uma.es (A.M.B.-M.); marilina@uma.es (M.G.-A.); teresatellez@uma.es (T.T.); 2Research Network on Chronic Diseases, Primary Care and Health Promotion (RICAPPS), Carlos III Health Institute (Instituto de Salud Carlos III). Av. De Monforte de Lemos, 5, 28029 Madrid, Spain; 3Málaga Biomedical Research Institute (Instituto de Investigación Biomédica de Málaga, IBIMA), Calle Doctor Miguel Díaz Recio, 28, 29010 Malaga, Spain; 4Research Unit, Hospital Costa del Sol, Autovía A-7, km 187, 29603 Marbella, Spain

**Keywords:** breast cancer, immunotherapy, protein kinase, inhibitor, antibody monoclonal, combination therapies

## Abstract

**Simple Summary:**

Recent research has revealed a possible synergy between kinase inhibitors and immunotherapy in the treatment of breast cancer. Kinase inhibitors can modulate the tumor microenvironment, making it more receptive to the infiltration and activation of immune cells. This modulation increases the efficacy of subsequent immunotherapeutic interventions, such as immune checkpoint inhibitors. Moreover, targeting specific kinase pathways may reverse the immune evasion mechanisms employed by tumor cells, making them more susceptible to immune recognition and destruction. This dual approach promises to improve response rates and outcomes in breast cancer patients, particularly in kinase-disrupted subtypes. The convergence of kinase inhibitors and immunotherapy represents an exciting frontier in breast cancer treatment. Thus, in this review, we will examine these combinations and their potential impact on the efficacy of responses to these treatments.

**Abstract:**

Cancer is one of the world’s most significant health problems today. Currently, breast cancer has globally surpassed lung cancer as the most commonly diagnosed cancer in women. In 2020, an estimated 2,261,419 new cases were diagnosed in women worldwide. Therefore, there is a need to understand the processes that can help us better treat this disease. In recent years, research in the fight against cancer has often been based on two treatment modalities. One of them is the use of protein kinase inhibitors, which have been instrumental in the development of new therapeutic strategies. Another crucial route is the use of immunotherapy, which has been touted as a great promise for cancer treatment. Protein kinase alterations can interfere with the effectiveness of other treatments, such as immunotherapy. In this review, we will analyze the role played by protein kinase alterations in breast cancer and their possible impact on the effectiveness of the response to immunotherapy treatments.

## 1. Introduction

Cancer is one of the major health problems in the world today and is one of the leading causes of premature death, second only to cardiovascular disease [1]. According to the World Health Organization, by 2040, 28.9 million new cases of cancer are expected to be diagnosed worldwide, and 16.2 million cancer-related deaths will occur annually [2]. Currently, breast cancer in women has globally surpassed lung cancer as the most commonly diagnosed cancer. In 2020, an estimated 2,261,419 new cases were diagnosed in women worldwide. Additionally, the incidence and mortality rates of this disease are 11.7% and 6.9%, respectively [3]. Therefore, both are expected to decrease once the disease is diagnosed before it progresses to metastasis.

Breast cancer presents many risk factors, making it impossible to have a homogeneous pattern for this disease [4]. Hereditary breast cancer accounts for 20–25% of the total, and 5–10% of all breast cancers are caused by mutations in *BRCA1/2* [5,6]. Similarly, mutations affecting *TP53* are also associated with triple-negative breast cancer (TNBC) [7].

Traditionally, breast cancer has been classified at the molecular level into three subtypes [8] (see Table 1).

Although early diagnosis significantly increases survival rates, early detection by screening programs has been associated with several disadvantages, such as high costs, overdiagnosis rates, risks from ionizing radiation, or recommendations for false-positive biopsies [13]. However, improved programs aimed at mammography interpretation, breast self-examination, clinical breast examination, digital breast tomosynthesis, ultrasound, magnetic resonance imaging, and oncogene identification are the main tests for early diagnosis, screening, and prevention of risk factors, as well as for timely treatment to reduce breast cancer morbidity [13,14]. Additionally, adjuvant chemotherapy has positively helped to improve breast cancer patients’ prognosis, overall survival, disease-free survival [15], and breast cancer-related mortality rates [16]. However, despite the development of screening programs, diagnosis, and treatment, approximately 6–10% of women will present with metastatic disease at the time of diagnosis, and 30% of breast cancer survivors will eventually progress to a metastatic and usually fatal disease [16].

The aim of this review is to analyze the role of inhibitors of altered protein kinases in breast cancer and their effectiveness in combination with immunotherapy as a possible pathway for therapeutic improvement in this pathology.

## 2. Protein Kinases

Protein kinases and phosphatases are part of a wide family of enzymes that regulate protein activity through the process of phosphorylation/dephosphorylation, one of the most common and relevant post-translational modifications. In humans, more than 500 enzymes carry out the reversible transfer of phosphate groups (PO4), diphosphate, nucleotidyl residues, and other groups to a receptor molecule [17,18]. These enzymes form the human kinome, and approximately 1.7% of all human genes encode them [19]. In recent years, different classification systems have been developed to better understand the human kinome. Traditionally, classification has been based on cellular location. On the one hand, there are transmembrane receptor kinases, which are composed of a ligand-binding extracellular domain and a catalytic intracellular kinase domain. On the other hand, there are non-receptor kinases, which are deprived of transmembrane domains and found in the cytosol, nucleus, or associated with the inner part of the cytoplasmic membrane. Additionally, these proteins are included in Class 2.7-Transferring phosphorus-containing groups of the Enzyme List made by the Nomenclature Committee of the International Union of Biochemistry and Molecular Biology [18], and they are related to the regulation of cellular activities [20]. Another classification, made by The Protein Kinase Complement of the Human Genome, divides them according to sequence comparisons of their catalytic domains [19].

These enzymes play a crucial role in cellular mechanisms such as proliferation, differentiation, and apoptosis [20]. Malfunctioning of these enzymes has been reported in the literature to be associated with various characteristics of cancer, including proliferation, survival, angiogenesis, resistance to treatment, and evasion of anti-tumor immune responses [21,22]. Consequently, certain protein kinases are classified as oncogenic because their transforming activity can dictate the survival and proliferation of tumor cells [23]. See Figure 1. Due to these reasons, protein kinases have become one of the primary pharmacological targets in our time.

## 3. Target Protein Kinases in Breast Cancer

The regulation of target proteins via phosphorylation by kinases is meticulously controlled, and any disruption to this regulatory process can result in the development of pathological conditions. Numerous kinases have been observed to exhibit dysregulation in a variety of cancer types [25]. Several mechanisms can contribute to the dysregulation of kinases, thereby amplifying their oncogenic potential. These mechanisms encompass overexpression, subcellular relocalization, fusion events, point mutations, or the aberrant modulation of upstream signaling pathways [26,27,28]. Studying the function of kinases has not only contributed to the progress of cancer biology but has also ushered in the era of “targeted therapy” and “personalized medicine” for cancer, bringing about a fundamental change in how cancer is treated. Below, we outline the primary protein kinase targets in breast cancer.

EGFR

The human epidermal growth factor receptor (HER) family of receptors occupies a central position in the development of various human cancers. This family consists of four principal members, namely HER1, HER2, HER3, and HER4, which are also referred to as ErbB1, ErbB2, ErbB3, and ErbB4, respectively.

HER receptors are initially found as single monomers on the cell surface. When ligands bind to their extracellular domains, the HER proteins undergo a process of dimerization, where two receptor molecules come together, and subsequently, transphosphorylation occurs within their intracellular domains. This dimerization can manifest as homodimerization, involving two identical receptors, or heterodimerization, involving different HER receptors. This event results in the autophosphorylation of tyrosine residues within the cytoplasmic domains of the receptors. This autophosphorylation then initiates a variety of signaling pathways, with a primary focus on the activation of mitogen-activated protein kinase (MAPK), phosphatidylinositol-4,5-bisphosphate 3-kinase (PI3K), and protein kinase C (PKC). These pathways lead to cell proliferation, survival, differentiation, angiogenesis, and invasion [29,30]. Amplification of the *HER2* gene (ErbB2) and corresponding overexpression of the HER2 receptor occurs in approximately 20–25% of breast tumors and is associated with a worse prognosis, shorter disease-free, and overall survival in breast cancer [31].

PI3K/Akt/mTOR

When an extracellular ligand activates the specific receptor, it triggers the activation of PI3K (phosphoinositide 3-kinase). Activated PI3K initiates the phosphorylation of PIP2, yielding PIP3, which in turn recruits two protein kinases to the plasma membrane: AKT and PDK1. Once translocated to the cell membrane, AKT is subject to phosphorylation by mTORC2. Activated AKT, in turn, phosphorylates target proteins on the cell membrane before disengaging from it and proceeding to phosphorylate other target proteins in the cytosol and the cell nucleus. This leads to a signaling pathway of paramount importance in various cellular processes, including cell metabolism, growth, proliferation, apoptosis, and angiogenesis [32].

Phosphatidylinositol 3-kinase (PI3K)

PI3K is an enzyme anchored to the plasma membrane, and it can be activated by both receptor tyrosine kinases (RTKs) and G protein-coupled receptors (GPCRs). GPCRs represent the largest category of cell surface receptors, characterized by a single transmembrane polypeptide chain that utilizes G-proteins to transmit signals to the cytoplasm [32,33,34,35]. On the other hand, RTKs are also a large family of plasma membrane receptors that possess intrinsic protein kinase activity [35,36,37].

Akt

The serine-threonine protein kinase AKT stands as the principal molecule downstream of the PI3K signaling pathway. Upon PI3K activation, it catalyzes the phosphorylation of PIP2 to produce PIP3, thereby recruiting AKT to the plasma membrane. The activation of AKT hinges on the phosphorylation of Thr308 and Ser473 [38,39]. Activated AKT plays a pivotal role in governing the cell cycle, growth, proliferation, and energy metabolism [40]. AKT possesses a diverse array of substrates, encompassing transcription factors, cell cycle progression inhibitors, protein kinases, GTPase activating proteins, and apoptosis inducers [38,41,42].

mTOR

TOR, a substantial protein kinase, assumes a dual role within cells by forming two distinct multiprotein complexes: mTORC1 and mTORC2.

mTORC1 is composed of mTOR, the Raptor protein, and mLST8. mTORC1 exhibits sensitivity to rapamycin and emerges as a key player in fostering cell growth and survival. It achieves this by promoting nutrient assimilation and metabolic processes, orchestrating ribosome production and protein synthesis, and concomitantly restraining protein degradation [32,43]. The activation of mTORC1 can be orchestrated through diverse pathways, with the PI3P/AKT pathway, primarily galvanized by extracellular growth factors and nutrient stimuli, occupying a prominent role.

PTEN

PTEN (phosphatase and tensin homolog) is an enzyme specific to PIP3 that catalyzes the dephosphorylation of PIP3 molecules, converting them into PIP2 molecules. PIP2 does not bind to AKT, preventing its incorporation into the cell membrane. Consequently, AKT cannot be phosphorylated by mTORC2 at Ser473, and the conformation of AKT remains unchanged. Phosphorylation at Thr308 by PDK1 is also inhibited, preventing AKT activation and suppressing the PI3K/AKT/mTOR signaling pathway [44,45]. Therefore, PTEN functions as a tumor suppressor by inhibiting cell proliferation. In many cancerous tumors, mutations in the *PTEN* gene result in altered PTEN, diminishing its capacity to impede the PIP3/AKT/mTOR pathway [46]. This results in an increase in plasma levels of PIP3 and continuous stimulation of AKT activity [47].

PDK

PDK phosphorylates PDHE1, which inhibits the PDC and redirects pyruvate metabolism to lactate [48]. There are four isoforms of PDK (PDK1, PDK2, PDK3, and PDK4) that have different biochemical characteristics, tissue-specific expression patterns, and functions [49,50,51,52,53]. Alterations in PDKs or inhibition of PDCs by PDKs are associated with cancer; therefore, inhibiting PDKs is an ideal option for treating this disease [54,55,56,57,58,59,60,61,62,63,64].

PDPK1

PDPK1 (3-phosphoinositide-dependent protein kinase-1) is a master kinase that plays a crucial role in the activation of AKT/PKB. Its main function is in signaling pathways activated by various growth factors and hormones, including insulin signaling [65]. PDPK1 consists of an N-terminal kinase domain and a C-terminal pleckstrin homology (PH) domain that senses phosphoinositide metabolites generated by PI3K, specifically phosphatidylinositol (3, 4, 5)-trisphosphate [66]. PDPK1 phosphorylates AKT and substantially increases Akt activity [67].

MAPK

MAPKs (mitogen-activated protein kinases) are a family of serine/threonine kinases that are activated by growth and stress factors. These proteins play a key role in intracellular signal transduction, allowing the cell to integrate different extracellular stimuli. Thus, MAPKs regulate processes such as mitosis, changes in gene expression patterns, metabolism, and programmed cell death, allowing cells to survive, proliferate, induce apoptosis, and interact with multiple cell types. All these processes are involved in the correct development of the organism, as well as its homeostasis, with implications for cancer and its therapy [68].

AURK

Aurora kinases (AURKs) are a family of serine/threonine kinases that are indispensable kinases in the process of cell division by regulating mitosis, and they are also involved in the regulation of meiosis. They are divided into Aurora A (AURKA), Aurora B (AURKB), and Aurora C (AURKC) [69]. Aurora kinases have a homologous structure consisting of an N-terminal domain, a protein kinase domain, and a C-terminal domain [70,71,72,73]. In various types of human cancers, such as breast cancer, overexpression or amplification of AURK is observed [74,75,76,77].

CDK

Cyclin-dependent protein kinases (CDKs) are crucial proteins in the regulation of the cell cycle. When activated, these enzymes phosphorylate Ser or Thr residues of their target proteins. Dysregulation of the cell cycle is a hallmark of cancer [78], with the development of numerous neoplasms determined by differential expression and mutations in *CDKs* [79]. The CDK family is divided into two groups. On one hand, CDK1, CDK2, (CDK4, and CDK6) are responsible for cell cycle progression. On the other hand, there are (TA-CDKs) responsible for carrying out gene transcription [79,80].

PLK

The polo-like kinases (PLKs) comprise a family of serine-threonine kinases. They possess a kinase domain at the N-terminal domain, and at the C-terminal end, they have one or two polo-box domains responsible for binding phosphopeptides [81]. The PLK family is composed of four members, of which PLK1 is the most well-characterized. During cell cycle progression, the expression level of PLK1 changes, reaching a maximum level in M-phase. Due to its role in protein binding and phosphorylation at various targets, PLK1 is essential in the vast majority of mitotic steps [82,83,84]. Overexpression of this enzyme suppresses mitotic checkpoints, resulting in immature cell division where normal chromosome alignment and segregation are absent. This leads to chromosomal instability and aneuploidy, which are characteristics of cancer [85]. Therefore, PLK1 is overexpressed in different types of cancer, such as colon [86], stomach [87], pancreatic [88], head and neck [89], ovarian [90], and breast [91]. Poor prognosis in cancer is often associated with PLK1 overexpression [92].

SK

Sphingosine kinases (SphK, SK) 1 and 2 are lipid kinases that catalyze the formation of sphingosine-1 phosphate (S1P), a potent signaling molecule with a wide range of cellular effects such as cell survival, proliferation, angiogenesis, differentiation, migration, and immune function, among others. There are several biological functions of the SphK1 and SphK2 isoforms, with each isoenzyme having multiple isoforms that differ only at the N-terminal end [93,94,95,96]. SphK1 and two have been shown to be up-regulated in tumor processes, and their ablation or genetic inhibition has been shown to slow tumor growth and sensitize cancer cells to chemotherapeutic agents [97,98,99]. SphK1 is implicated in breast cancers, representing a valid therapeutic target [100]. Therefore, the study of human SphK isozymes provides a better understanding of cancer progression, metastasis, and drug resistance [101].

PTK

Protein tyrosine kinases (PTKs) constitute a family of enzymes responsible for transferring ATP to the tyrosine residues on specific target proteins [102]. Tyrosine phosphorylation serves pivotal roles in a multitude of cellular processes, encompassing cell proliferation, differentiation, protein synthesis, cell cycle regulation, embryonic development, cell migration, and apoptosis [19,103,104,105]. Excessive expression of protein tyrosine kinases is linked to a range of diseases, including diabetes, neurodegenerative disorders, and cancer [29,106,107,108,109,110,111]. Consequently, extensive scientific literature attests that a number of human tumors, such as non-small cell lung cancer, squamous cell carcinoma of the head and neck, glioblastoma, pancreatic cancer, ovarian cancer, breast cancer, and prostate cancer, are intricately associated with the overactivation of the human epidermal growth factor receptor 2 (Her2), which belongs to the ErbB (erythroblastic oncogene *B*) protein tyrosine kinase subfamily [30,108,112,113].

Table 2 lists the main altered protein kinases, as described above, and the breast cancer subtype in which they are expressed.

## 4. Kinase Inhibitors as Targets in Breast Cancer

Tyrosine kinase inhibitors (TKIs) are a group of small molecule drugs that promote apoptosis and inhibit cancer cell proliferation. They can act on extracellular surface receptors as well as penetrate through the cell membrane and interact with intracellular targets. Typically, they are designed to interfere with the enzymatic activity of the target protein [114].

The inhibitor binds competitively to the intracellular adenosine triphosphate (ATP)-binding domains of the EGFR family due to the homologous structure of ATP. This results in the inhibition of tyrosine kinase phosphorylation, which subsequently blocks downstream signaling [115]. These inhibitors have several advantages, such as oral administration, multiple targets, and lower cardiotoxicity compared to intravenous monoclonal antibodies. Additionally, they can permeate the blood-brain barrier (BBB), which gives them greater efficacy over monoclonal antibodies in the treatment of certain cancers, such as those with brain metastasis [116].

### 4.1. EGFR Inhibitors

Anti-EGFR drugs inhibiting tyrosine kinase currently used in clinical practice include lapatinib, neratinib, tucatinib, and pyrotinib, which are currently approved, mostly in combination with other agents.

Lapatinib (Tykerb^®^, GlaxoSmithKline (GSK), Brentford, UK) is a TKI (tyrosine kinase inhibitor) with the ability to block HER1 and HER2 reversibly. Lapatinib restricts HER1 and HER2 phosphorylation by reversibly and competitively inhibiting the ATP binding sites of intracellular kinase regions. Subsequently, it disrupts downstream signaling, namely Raf, AKT, ERK, and PLC γ, resulting in significant efficacy in inducing apoptosis and directing the development and migration of HER2-overexpressing cancer cells [117,118]. However, the level of HER1 expression is irrelevant to the antineoplastic effect of lapatinib in HER2-overexpressing breast cancer cells [119]. Lapatinib has been administered in combination with capecitabine to treat patients with HER2-overexpressing breast cancer who have previously undergone anthracycline, taxane, and trastuzumab therapies since its approval in 2007. Furthermore, in 2010, it gained approval for use in conjunction with letrozole in postmenopausal patients with metastatic breast cancer expressing hormone receptors and HER2 (NCT00073528) [120].

In contrast to lapatinib, neratinib (HKI-272), trade name Nerlynx^®^ (Puma Biotechnology, Los Ángeles, CA, USA), is an irreversible TKI of HER1, HER2, and HER4. Neratinib inhibits phosphorylation of the ErbB family, as well as downstream pathways, including ERK and Akt [121]. Inhibition of downstream signal transduction causes reduced phosphorylated retinoblastoma protein (pRB) and expression of cyclin D1 and p27 upstream level, which arrests the G1-S phase transition, eventually resulting in the negative regulation of cell proliferation [122]. This mechanism of action makes neratinib, in 2017, a suitable therapy for the treatment of patients with early-stage HER2+ breast cancer, based on the results of the ExteNET clinical trial (NCT00878709), which was subsequently switched to combination therapy with capecitabine, as the results significantly outperformed lapatinib-capecitabine in terms of progression-free survival (PFS) (NALA) [123,124]. The most frequent grade 3 or 4 adverse events (AEs) were diarrhea, nausea, hand-foot syndrome, rash, and fatigue.

Tucatinib belongs to a new generation of tyrosine kinase inhibitors that exhibit higher specificity for HER2 and lower specificity for EGFR. Tucatinib, marketed as Tukysa^®^ (Seattle Genetics, Bothell, WA, USA), is the first drug to demonstrate improved overall survival and progression-free survival in previously treated HER2-positive metastatic breast cancer patients, regardless of the presence of brain metastases. It is typically administered in combination with trastuzumab and the chemotherapy drug capecitabine. The FDA approved its use in April 2020 for the treatment of advanced metastatic or unresectable breast cancer. Preclinical studies and phase I clinical trials have shown promising results for tucatinib as a single-agent, and its efficacy was further enhanced when combined with chemotherapy or trastuzumab [125,126,127,128,129,130,131,132,133,134,135,136,137,138,139]. Understanding the signaling mechanism reveals that unlike other dual EGFR and HER2 inhibitors such as lapatinib and neratinib, tucatinib specifically and reversibly inhibits the protein tyrosine kinase (PTK) activity of HER2 while exerting minimal inhibition on EGFR [129,130,140].

In terms of therapeutic responses, the results obtained from the HER2CLIMB study demonstrate that the combination of tucatinib with trastuzumab and capecitabine shows higher effectiveness compared to trastuzumab and capecitabine alone in treating individuals diagnosed with HER2-positive breast cancer. This combination therapy can be employed to reduce the likelihood of disease progression or to enhance survival advantages in patients with HER2-positive metastatic breast cancer, regardless of the presence of brain metastases [134,135]. Refer to Table 3 for further details.

Pyrotinib (SHR-1258) dimaleate, marketed under the trade name Irene^®^, is an irreversible pan-ErbB receptor TKI developed by Jiangsu Hengrui Pharma. It exhibits activity against HER1, HER2, and HER4. In 2018, it was approved by the State Drug Administration of China for combination therapy with capecitabine in patients with advanced or metastatic HER2-positive breast cancer who had previously received trastuzumab and taxane [139]. This drug functions by inhibiting the formation of homologous/heterodimeric complexes within the HER family and suppressing autophosphorylation, effectively blocking the activation of signaling pathways such as RAS/RAF/MEK/MAPK, PI3K/AKT, and the G1-phase tumor cell cycle. This inhibition restricts tumor progression. Preliminary findings have indicated that pyrotinib, combined with capecitabine, significantly improves progression-free survival compared to lapatinib plus capecitabine while exhibiting manageable toxicity. Therefore, it may be considered a viable alternative treatment option for patients with HER2-positive metastatic breast cancer following trastuzumab and chemotherapy [141]. The study is registered under the identifier NCT03691051 and is currently ongoing, but recruitment has been terminated.

Other TKIs are being evaluated, either as monotherapy or in combination with other drugs, in ongoing phase I and II clinical trials, with promising results. Poziotinib, an irreversible pan-HER TKI, has demonstrated significant anti-cancer properties in both HER2+ cancer cell lines [142] and patients participating in a phase 1 clinical trial [143]. Interestingly, it has been observed that poziotinib enhances the expression of HER2 and exhibits a synergistic effect when used in conjunction with T-DM1 [144]. In a phase 2 clinical trial, poziotinib was administered to patients with heavily pretreated HER2+ metastatic breast cancer who had previously undergone two HER2-targeted therapies, resulting in encouraging anti-cancer activity [145]. The most commonly reported side effects included diarrhea and stomatitis. Currently, there is an ongoing clinical trial (NCT03429101) investigating the combination of poziotinib and T-DM1.

Epertinib, also known as S-222611, is an inhibitor of pan-HER that acts reversibly and has demonstrated enhanced efficacy in both in vitro and in vivo settings [146]. It has displayed promising outcomes in a HER2+ brain metastasis model [147]. In a phase 1 clinical trial, epertinib was tested in patients with solid tumors, including HER2+ tumors, revealing effective antitumor activity, particularly in brain metastases, along with good tolerability and safety [148].

In preclinical studies involving breast cancer and brain metastases, the reversible and selective HER2 inhibitor DZD1516 has demonstrated a significant reduction in tumor size [149]. Currently, a phase 1 clinical trial is underway to evaluate the combined impact of DZD1516 with trastuzumab and capecitabine or T-DM1 (NCT04509596). Preliminary findings suggest that DZD1516 effectively penetrates the blood-brain barrier and exhibits a favorable safety profile. Interestingly, diarrhea has not been reported as an adverse event in this investigation [150].

### 4.2. Cyclin-Dependent Kinase 4/6 Inhibitors: CDK4 and CDK6

The introduction of inhibitors targeting cyclin-dependent kinase (CDK) 4 and 6 has brought about a significant transformation in the treatment landscape of hormone receptor-positive (HR+) and HER2-negative (HER2−) metastatic breast cancer. This development can be argued as one of the most significant advancements in breast oncology over the past two decades. Three medications, namely palbociclib, ribociclib, and abemaciclib, have received approval for use in combination with aromatase inhibitors and fulvestrant for patients with HR+, HER2− metastatic breast cancer [151,152,153,154,155]. Additionally, abemaciclib has gained approval for use as a monotherapy.

Palbociclib (Ibrance^®^, Pfizer, New York, NY, USA) was approved by the FDA in 2015 for the treatment of postmenopausal women with hormone HR+, HER2− metastatic advanced breast cancer in combination with letrozole following the results of a phase III study, PALOMA-2. The results showed that the addition of palbociclib to letrozole significantly improved progression-free survival compared to letrozole alone. Not only has it been tested in combination with letrozole, but another study tested safety and efficacy in combination with fulvestrant, the phase III PALOMA-3 study. The results of this study also demonstrated a significant improvement in progression-free survival with the addition of palbociclib to fulvestrant compared to fulvestrant alone. An advantage of palbociclib is that it has few side effects and may reduce the number of white blood cells, but it does not appear to increase the risk of serious infections associated with a low white blood cell count [156,157].

Ribociclib (Kisqali^®^, Novartis, Basel, Switzerland) was approved by the FDA in 2015 based on the results of the Phase III MONALEESA-2 study (NCT01958021) of postmenopausal women with HR+, HER2− advanced or mBC who had not received prior treatment for advanced-stage cancer. Subsequently, the MONALEESA-2 study findings showed a notable extension in progression-free survival (PFS) and a well-tolerated side effect profile when using ribociclib alongside letrozole as a first-line treatment [158]. In contrast to palbociclib, the group of premenopausal and perimenopausal patients with metastatic breast cancer is treated with ribociclib in combination with fulvestrant following the results of the MONALEESA-3 and MONALEESA-7 trials and their 2018 FDA approval [159,160]. In terms of adverse effects, ribociclib altered the heart rhythm in 3% of patients, while palbociclib did not have this effect. The most common adverse events for patients are listed in Table 3.

Abemaciclib (Verzenio^®^, Eli Lilly and Company, Indianapolis, IN, USA) was the third inhibitor to be approved by the FDA in October 2017 as a first-line aromatase inhibitor (AI) treatment for postmenopausal women with ER+, HER2− mBC. Results from the phase III clinical trial, MONARCH 3 (NCT02246621), contributed to the approval of this drug. These showed that in addition to having longer progression-free survival, women treated with abemaciclib and an aromatase inhibitor experienced some reduction in the size of their tumors [161]. It can also be used in conjunction with fulvestrant if the cancer has worsened with other hormonal therapies. Although all three drugs are similar in terms of mechanism of action and efficacy, there are some structural differences between them. Abemaciclib is structurally different from other CDK 4 and 6 inhibitors, such as ribociclib and palbociclib, and is much more potent against cyclin D1/CDK4 and D3/CDK6 in enzyme assays. Palbociclib and ribociclib have similar cores and chains (except for an extreme carbonyl group), while abemaciclib has a different chain, giving the molecule versatility with important functional consequences. Toxicity profiles also differed, with about 20% of women in the study treated with abemaciclib having to discontinue treatment due to side effects and more than 40% having to reduce the drug dose. The most frequent serious side effect for those receiving the drug was diarrhea, which occurred in about 80% of patients. Mild fatigue and nausea were also common in women treated with abemaciclib.

### 4.3. PI3K Inhibitors

The next target is at the intracellular signaling level. It is common in breast cancer for the PI3K pathway to be aberrantly activated, leading to uncontrolled tumor cell growth and drug resistance. This is due to mutations in *PIK3CA*, the gene encoding the p110α catalytic subunit of PI3K, and occurs frequently in HR+ breast cancer, with an incidence of approximately 40% of cases [162]. Multiple categories of drugs are aimed at the PI3K network, including pan-PI3K inhibitors, isoform-specific PI3K inhibitors, AKT inhibitors, analogs of rapamycin or mTOR inhibitors, and compounds that target both PI3K and mTOR simultaneously [163].

#### 4.3.1. Pan-PI3K Inhibitors

Pan-PI3K inhibitors inhibit the kinase activity of all four class I PI3K isoforms: α, β, γ, and δ. Buparlisib is an oral pan-PI3K inhibitor indicated for patients with HR+, HER2− advanced, or mBC [164]. Several BELLE trials (BELLE-2, BELLE-3) evaluated the efficacy and safety of this inhibitor in combination with fulvestrant, resulting in a modest improvement in progression-free survival (PFS) but with many adverse effects. It was also evaluated in combination with paclitaxel (phase II/III BELLE-4 trial). The addition of buparlisib to paclitaxel did not improve PFS, and the trial was stopped for futility at the end of phase II [165,166].

#### 4.3.2. Isoform-Specific PI3K Inhibitor

Isoform-specific PI3K inhibitors may offer the opportunity for higher doses and potentially greater efficacy with less toxicity compared with pan-PI3K inhibitors [11]. Selective inhibitors include alpelisib and taselisib. Alpelisib (Piqray^®^, Novartis, Basel, Switzerland) is a kinase inhibitor that was developed for use in combination with endocrine therapy, specifically fulvestrant, to help overcome endocrine resistance in HR+/HER2− advanced breast cancer. It targets mutations in *PIK3CA*, which is the gene encoding the p110α catalytic subunit of phosphatidylinositol 3-kinase (PI3K).

Results from the phase III SOLAR-1 trial demonstrated that alpelisib plus fulvestrant nearly doubled the median PFS and overall response rate compared with fulvestrant alone. As a result, alpelisib was approved by the FDA on 24 May 2019 for postmenopausal women and men with HR+/HER2−, advanced, or mBC with a *PIK3CA* mutation after progression or an endocrine regimen [167]. The most frequent grade 3–4 adverse events observed in patients receiving alpelisib included hyperglycemia, severe skin reaction, and diarrhea. Additional adverse reactions include severe hypersensitivity and pneumonitis, which are listed in Table 3.

Taselisib has been specifically investigated in patients with HR+, HER2− early breast cancer. The combination of taselisib and letrozole before surgery improved outcomes according to the LORELEI trial (NCT02273973) [167]. However, when tested in patients with advanced breast cancer, the combination of taselisib plus fulvestrant has no clinical utility, given its safety profile and modest clinical benefit (SANDPIPER trial) [168].

### 4.4. mTOR Inhibitor

An additional pathway commonly disrupted in cancer is the PAM pathway (PI3K/AKT/mTOR), which is implicated in over 70% of breast cancer cases. In HER2+ breast cancers, the oncogene *PIK3CA2* is frequently mutated. Consequently, numerous clinical trials have concentrated on targeting this pivotal signaling pathway, which plays a role in critical cellular functions. While research has demonstrated substantial potential for multiple PI3K and AKT inhibitors, the undesirable side effects associated with these compounds have restricted clinical trials to the use of only one mTOR inhibitor, namely everolimus [169,170].

Everolimus is an oral analog (analog of rapamycin) used for the treatment of postmenopausal women with HR+, HER2− advanced breast cancer in combination with exemestane and after letrozole or anastrozole treatment failure. Everolimus was approved by the FDA on 20 July 2012 [169,170,171,172]. It is administered in combination with trastuzumab and vinolrebine in patients with HER2+ and refractory breast cancer, following the results of the BOLERO-3 clinical trial evaluating its efficacy [173]. Everolimus has also exhibited potential when used in conjunction with trastuzumab and paclitaxel for HER2+ advanced breast cancers [174]. The MANTA trial has provided evidence that the combination of everolimus and fulvestrant substantially enhances progression-free survival [175]. Additional research studies have indicated that everolimus enhances the efficacy of letrozole by impeding the mammary cell cycle and promoting apoptosis [176].

Dual mTOR inhibitors have shown promising results in the suppression of tumor growth by targeting both mTORC1 and mTORC2. However, while the initial generation of mTORC1 inhibitors has gained approval for clinical trials, dual inhibitors using cell culture and animal models are still under investigation. Vistusertib (AZD2014) is an example of a dual inhibitor that affects both mTORC1 and mTORC2, demonstrating more comprehensive growth inhibition and cell death in both in vitro and in vivo settings when compared to everolimus. Its inhibitory effect on mTORC1 is heightened, and it additionally inhibits mTORC2, particularly in models of ER+ breast cancer, and induces rapid tumor regression in preclinical models [175]. Vistusertib is being evaluated in an international, multicenter, Phase I/II study of the combination of AZD2014 and palbociclib in the context of fulvestrant in postmenopausal women with locally advanced/metastatic ER+ cancer (PASTOR, NCT02599714). Part C is a randomized, double-blind, Phase II trial comparing the triplet (AZD2014, palbociclib, and fulvestrant) with the doublet (palbociclib and fulvestrant). Patients will be assessed by measuring progression-free survival. Table 3 and Figure 2 provide more information on the study [177].

## 5. Protein Kinase-Targeted Immunotherapy in Breast Cancer

Immunotherapy is a type of treatment that stimulates the body’s natural defenses to fight cancer. Substances produced by the body or in a laboratory are used to improve the functioning of the immune system, thus destroying cancer cells.

### 5.1. Immunotherapy Targeting EGFR

The EGFR-targeted antibodies that have been tested so far are cetuximab and panitumumab. Neither of these antibodies has shown effectiveness in treating TNBC breast cancer. Cetuximab is a recombinant human/mouse chimeric IgG1 monoclonal antibody that specifically binds to EGFR on both normal and tumor cells, competitively inhibiting the binding of epidermal growth factor (EGF). Results from the phase II TBCRC 001 trial, which evaluated the use of cetuximab alone or in combination with carboplatin in patients with metastatic TNBC, indicated that less than 20% of patients responded positively [178]. The efficacy of a similar antibody, panitumumab, is currently being assessed in a randomized phase 2 trial investigating carboplatin with and without panitumumab in TNBC patients (NCT02876107). The lack of effectiveness of these antibodies is likely attributed to the fact that they are designed to block the activity of the epidermal growth factor receptor (EGFR), which is overexpressed in some cancer cells, a characteristic not observed in TNBC breast cancer. [179]

#### 5.1.1. Immunotherapy against HER2

The monoclonal antibodies used against this receptor can be administered as monotherapy or, in combination with other therapies, or in the form of antibody-drug conjugates (ADCs).

##### Monoclonal Antibodies

The following monoclonal antibodies have been studied targeting this receptor:

Trastuzumab (Herceptin^®^, Roche, Basel, Switzerland): It is a recombinant humanized IgG1 monoclonal antibody that specifically binds to the extracellular domain of HER2. This binding inhibits the signaling pathways that promote cell growth and proliferation upon receptor activation. Trastuzumab also facilitates antibody-dependent cell-mediated cytotoxicity (ADCC) by recruiting immune cells, such as natural killer (NK) cells, to the tumor microenvironment. It was the first HER2-targeted therapy to receive FDA approval in 1998 and is the established treatment for patients with HER2+ breast cancer [180].

Pertuzumab (Perjeta^®^, Roche, Basel, Switzerland) is also a humanized IgG1 directed against HER2 for the treatment of HER2+ breast cancer. It binds specifically to a different region of the HER2 receptor compared to trastuzumab. Pertuzumab binds to the extracellular domain of HER2, specifically targeting the dimerization domain. This binding prevents HER2 from forming heterodimers with other HER family receptors, such as HER3, which are necessary for the activation of downstream signaling pathways involved in cell growth and survival. By inhibiting HER2 heterodimerization, pertuzumab acts synergistically with other HER2-targeted therapies, such as trastuzumab, to provide a complete blockade of HER2 signaling and improve treatment outcomes for patients with HER2+ breast cancer. It is administered to patients with early metastatic breast cancer. It was approved by the FDA in 2012 [181].

PHESGO (Roche, Basel, Switzerland) is a combination of the above two monoclonal antibodies and hyaluronic acid used in patients with HER2+ early-stage and mBC. Unlike traditional treatments that require separate infusions of pertuzumab and trastuzumab, PHESGO is administered as a subcutaneous injection. The combination of pertuzumab and trastuzumab in PHESGO provides a more comprehensive targeted therapy against HER2. These monoclonal antibodies bind to the HER2 receptor on cancer cells and block signaling pathways that promote their growth and survival. The hyaluronic acid in PHESGO is used as a stabilizer to prolong the release of pertuzumab and trastuzumab in the body, allowing for more convenient and less invasive subcutaneous administration. This means patients can receive treatment at home or in a clinic without the need for prolonged intravenous infusions. PHESGO has been shown to be effective in clinical studies, both in the treatment of early-stage HER2+ breast cancer prior to surgery and in metastatic breast cancer. Results have shown a significant reduction in tumor size and improved survival outcomes. PHESGO was approved by the FDA in 2020 [182].

##### Fc-Optimized Antibody

Fc engineering aims to integrate the functions of antibodies, which encompass effector functions such as antibody-dependent cellular cytotoxicity (ADCC), antibody-dependent cellular phagocytosis (ADCP), and serum half-life regulation. An optimized Fc antibody used in the treatment of breast cancer is Margetuximab-cmkb [183].

Margetuximab-cmkb (Margenza™) is a chimeric IgG1κ monoclonal antibody engineered with an anti-human/mouse IgG1κ Fc region, targeting HER2. It has a unique modification in the Fc region compared to other HER2-targeted antibodies. This modification involves an amino acid substitution known as L234A/L235A, which reduces the interaction of the Fc region with immune system effector proteins. The intention behind this modification is to enhance the interaction with natural killer (NK) cells and stimulate a more effective immune response against HER2+ cancer cells. Margetuximab-cmkb received FDA approval in 2020 for the treatment of HER2+ mBC [184].

##### Antibody-Drug Conjugates (ADCs)

An antibody-drug conjugate (ADC) is a targeted therapy used in the treatment of breast cancer and other types of cancer. It consists of a specific monoclonal antibody linked to a cytotoxic drug [185]. The efficacy of these conjugates is contingent on the antibodies’ selectivity and the constrained number of molecules permitted to penetrate the cell. The potency of cytotoxic drugs employed in conjunction with ADCs often exceeds that of traditional chemotherapy [186].

The ADCs used in breast cancer include trastuzumab deruxtecan (T-DXd), trastuzumab emtansine (T-DM1), and trastuzumab duocarmazine. Trastuzumab deruxtecan (T-DXd) is an approved drug for the treatment of metastatic or locally advanced HER2+ breast cancer that has not responded to previous treatments or has progressed after initial therapy. It is marketed under the name Enhertu^®^ (Daiichi Sankyo, Tokyo, Japan, and AstraZeneca, London, UK) and combines the antibody trastuzumab with a cytotoxic drug called deruxtecan. Trastuzumab selectively binds to the HER2 receptor, while deruxtecan is a topoisomerase I inhibitor that interferes with DNA replication, causing selective damage and apoptosis in cancer cells by its ability to cross cell membranes [187]. It was approved by the FDA in 2022 [188]. Ongoing investigations are delving into the use of T-DXd in combination with various therapeutic approaches. Notably, T-DXd is being explored in conjunction with pertuzumab for the treatment of metastatic breast cancer (NCT04784715). Furthermore, the potential of T-DXd is being examined in combination with durvalumab (an anti-PD-L1 agent, NCT04538742) and tucatinib (a tyrosine kinase inhibitor targeting HER2, NCT04538742 and NCT04539938) in metastatic HER2+ breast cancer. Additionally, in the context of HER2-low breast cancer, T-DXd is under investigation in combination with a range of other pharmaceutical agents (NCT04556773) [189]. Encouraging initial results from T-DXd trials in HER2-low breast cancer have demonstrated significant efficacy and tolerability (NCT02564900) [190,191], leading to its inclusion in ASCO guidelines. The impressive success of T-DXd has prompted the initiation of multiple clinical trials aimed at assessing its potential synergies with immune checkpoint inhibitors, endocrine therapy, and chemotherapy, among other therapeutic modalities, in the treatment of low HER2 breast cancer (NCT04556773).

Trastuzumab emtansine (T-DM1), marketed as Kadcyla^®^ (Roche, Basel, Switzerland) is a conjugate of trastuzumab and a toxin called emtansine (DM1). DM1 is a derivative of maytansine that inhibits microtubules. Trastuzumab emtansine specifically binds to subdomain IV of the HER2 receptor and enters the cell through receptor-mediated endocytosis. Within the lysosomes, trastuzumab emtansine is degraded, releasing DM1, which then binds to microtubule tubulin and inhibits its function. This leads to cell cycle arrest and apoptosis. In terms of clinical efficacy, T-DM1 has demonstrated effectiveness in the treatment of HER2-positive breast cancer at various stages of the disease. It received FDA approval in 2013 for use in patients with HER2+ mBC who have previously received taxanes and/or trastuzumab for metastatic disease or whose cancer has recurred within six months of adjuvant treatment. Subsequently, the approval of T-DM1 has been expanded to include its use in early-stage HER2+ breast cancer following surgery [192,193]. The third combination antibody therapy includes trastuzumab duocarmazine. It consists of a humanized monoclonal antibody against immunoglobulin G1-kappa, conjugated to the prodrug seco-duocarmycin-hydroxib. Trastuzumab duocarmazine represents a novel therapeutic approach for the treatment of patients in three different oncology situations: those with HER2+ breast cancer resistant to prior T-DM1, those with HER2-low-expressing breast cancer who currently have no targeted therapy available, and other tumor types overexpressing HER2 [194,195,196].

##### Bispecific Antibodies and Bispecific Antibodies Based ADC’s

Currently, bispecific antibodies (BsAbs) are being used as a therapy for breast cancer. These are protein constructs based on naturally occurring mammalian antibody protein sequences that can bind to two or more different antigens simultaneously. Typically, one end of the BsAb targets an antigen on the effector cell, while the other end targets an antigen on tumor cells. There are many variations in the design and application of BsAbs that are relevant to breast cancer therapy and research. The first clinical trials were conducted in the 1990s [197]. A large number of bispecific antibodies against breast cancer are currently available and being studied in various clinical trials with promising results. The BsAbs under study include Zw25, ZW49, KN026, MCLA-128/zenocutuzumab, ISB-1302, HRE2Bi ATC, MBS301, M802, BCD-147, DF-1001, BTRC4017A, and IBI315. Of all these studies, we have highlighted those where results have already been obtained:Zanidatamab (ZW25) is a novel, humanized, bispecific monoclonal antibody that targets the juxtamembrane extracellular domain and the dimerization domain of HER2. In a phase I clinical trial (NCT02892123), zanidatamab demonstrated preliminary antitumor activity and had a tolerable safety profile when used alone or with chemotherapy in patients with pre-treated advanced HER2+ breast cancer. ZW25 has been well-tolerated with promising anti-tumor activity in patients with HER2-expressing carcinoma who progressed after standard-of-care therapy [198]. Currently, multiple trials are underway to assess the efficacy of ZW25 in various settings: ZW25 in early-stage breast cancer (NCT05035836) and its use in combination with palbociclib and fulvestrant for advanced breast cancer (NCT04224272). Additionally, ZW25 is being studied alongside an anti-CD47 agent in HER2+ solid tumors, encompassing both HER2-overexpressing and HER2-low breast cancers (NCT05027139).ZW49 is an antibody-drug conjugate (ADC) composed of a HER2-targeting bispecific antibody (which targets two non-overlapping HER2 epitopes) attached to a proprietary auristatin toxin with a protease-cleavable linker. In a phase I clinical trial (NCT03821233), ZW49 showed a manageable safety profile and encouraging single-agent antitumor activity in heavily pretreated patients with HER2-positive cancers [199].

Figure 3 and Table 4 show the main therapies described above and some of them in clinical trials.

### 5.2. Immunotherapy Against CDK4/6

CDK4/6 is a cell-localized protein kinase and, therefore, is inaccessible to antibody therapy. However, there is evidence of a direct interaction between CDK4/6 kinase inhibitors and immunomodulation, as CDK4 regulates PD-L1 protein stability. Hence, the concurrent application of kinase inhibitors along with anti-PD-1 immunotherapy amplifies tumor regression and augments the rates of overall survival in murine tumor models [200]. Presently, a number of clinical investigations are in progress to assess the advantages of employing these combination treatments, as delineated in Section 6.3.

## 6. Immunotherapy Against PD-1/PD-L1 Checkpoints in Breast Cancer

Programmed cell death 1 (PD-1) and programmed cell death ligand 1 (PD-L1) proteins are immune checkpoints. PD-1 (CD279) is a glycoprotein that is expressed on cytotoxic T cells, activated epithelial cells, and cancer cells [201,202]. PD-1 interacts with two ligands: programmed death ligand 1 (PD-L1) and programmed death ligand 2 (PD-L2) [203,204]. PD-L1, which is the primary ligand of PD-1, is constitutively expressed in normal epithelial, myeloid, lymphoid, and cancer cells [204]. PD-L2 is expressed in macrophages and some activated dendritic cells [205,206]. When PD-1 binds to PD-L1, it limits the action of the immune system and allows breast tumor cells to evade the immune response [207,208]. The principal effects of PD-1/PD-L1 signaling are co-stimulatory signaling and T-cell receptor blockade [209]. PD-1/PD-L1 inhibitors cause PD-L1 dimerization and dissociation of the complex [210,211].

### 6.1. PD-1 and PD-L1 Signaling in Breast Cancer

The interaction between PD-1 and its ligand PD-L1 leads to successive phosphorylations that result in spleen tyrosine kinase (Syk) and phosphatidylinositol 3-kinase (PI3K) phosphorylation, which inhibits T-cell activation and cytokine production. This subsequently leads to the depletion and apoptosis of tumor-specific T cells, promoting immune evasion of tumor cells [212,213]. PD-L1 expression on the surface of tumor cells has been characterized as an essential biomarker [214,215,216,217,218,219], specifically in HER2-positive breast cancer and triple-negative breast cancer [220]. Combinations of PD-1/PD-L1 inhibitors with different suppressors of molecules involved in intracellular signaling pathways, including kinase inhibitors, are currently being used to address multidrug resistance in breast cancer treatment [221,222].

### 6.2. Monotherapy Targeting PD-1 and PD-L1 Checkpoint Signaling in Breast Cancer

Drugs targeting the PD-1/PD-L1/PD-L2 signaling pathway block the interaction between PD-1 and its ligands by targeting either PD-1 (pembrolizumab and nivolumab) or PD-L1 (avelumab and atezolizumab).

Pembrolizumab is a humanized monoclonal antibody of the IgG4 isotype with a high affinity for PD-1, whose mechanism of action is based on blocking the interaction between the PD-1 receptor on T cells and the PD-L1 ligand on tumor cells. By blocking this interaction, pembrolizumab enables the immune system’s T cells to recognize and attack cancer cells. This can result in a more effective immune response against cancer and a reduction in tumor growth [223]. In terms of clinical efficacy, results from several clinical studies have demonstrated the positive antitumor activity of pembrolizumab as monotherapy in patients with PD-L1-positive advanced TNBC who have been previously treated with chemotherapy. These studies include the KEYNOTE trial (012-NCT01848834; 086-NCT02447003) [224,225]. The safety profile of pembrolizumab has also been found to be tolerable in patients with metastatic triple-negative breast cancer. A phase three clinical trial called KEYNOTE-355 was conducted to evaluate the combination of pembrolizumab with chemotherapy. This trial demonstrated a clinically significant antitumor effect. Unlike the IMPASSION-130 clinical trial (NCT02425891), which investigated the combination of atezolizumab and nab-paclitaxel, KEYNOTE-355 (NCT02819518) explored multiple therapeutic standards, including nab-paclitaxel, paclitaxel, or gemcitabine plus carboplatin, thereby expanding the range of treatment options.

Another monoclonal antibody, nivolumab, which blocks both the PD-1/PD-L1 and PD-1/PD-L2 interactions, is also being evaluated. Initial findings from the initial phase of the second-stage clinical trial TONIC reveal that nivolumab exhibits significant clinical effectiveness in individuals with metastatic TNBC. The research proposes that prior administration of immunomodulatory chemotherapeutic drugs, like cisplatin and doxorubicin, fosters a more conducive tumor microenvironment and results in a robust and long-lasting response to nivolumab [226].

Avelumab represents another immune checkpoint inhibitor. This monoclonal antibody of human IgG1 origin, directed towards PD-L1, effectively hinders the PD-1/PD-L1 pathway while leaving the PD-1/PD-L2 interaction unaffected [208,227]. What sets this agent apart is its secondary mode of action. In preclinical investigations, avelumab has exhibited an added advantage in the form of its capacity to elicit antibody-dependent cell-mediated cytotoxicity (ADCC), leading to the destruction of human cancer cells. The antitumor effect of avelumab has been evaluated in the JAVELIN study (NCT01772004) in advanced and metastatic breast cancer, revealing a favorable safety profile [228].

Atezolizumab is a humanized monoclonal antibody of the immunoglobulin G1 (IgG1) class that binds to the PD-L1 protein, aiding immune cells in the destruction of cancer cells [208,229]. These findings are based on the NCT01375842 trial conducted in patients with metastatic TNBC, which demonstrated that atezolizumab could provide patients with a stable and long-lasting response while maintaining an acceptable safety profile [230,231,232]. Currently, clinical trials are underway to assess the combination of monoclonal antibodies with kinase inhibitors and immune checkpoint inhibitors, aiming to enhance treatment response.

### 6.3. Combination of Anti-PD-1/PD-L1 Agents with Kinase Inhibitors and Other Therapies

With the aim of improving the anti-tumor response in breast cancer, the use of combination therapy is a major breakthrough [233,234]. Several trials currently underway are described below.

#### 6.3.1. Anti-PD-1 with Cyclin-Dependent Kinase 4/6 (CDK4/6) Inhibitors


**
*NCT04355858*
**


This study is a prospective, open-label, single-center phase II clinical trial with an umbrella design for patients with advanced HR+/HER2-resistant breast cancer. Seven precision treatment cohorts are targeted, focusing on NF1 mutation, gBRCA mutation, HER2 mutation, FDGFRb mutation, PAM pathway mutations, CD8 mutations, and AR mutations, provided that an epigenetic therapy cohort and a combined immunization cohort are initially established based on gene expression profiles and molecular pathways. The primary objective is to select valuable treatment cohorts and prepare for subsequent phase III randomized controlled clinical trials with a larger sample size. One of the arms, Arm-5F, evaluates the combination of SHR1210 (anti-PD-1 antibody) and SHR6390 (CDK4/6 inhibitor) with AI (aromatase inhibitor). Currently, in phase 2, the first results are expected in May 2023 [235].


**
*NCT02779751*
**


Phase 1b clinical trial assessing the combination of abemaciclib and pembrolizumab in patients with HER2-negative, hormone receptor-positive breast cancer. Within this trial, one branch of the study focuses on patients with metastatic breast cancer who are HER2-negative and hormone receptor-positive. In this arm, abemaciclib is administered orally every 12 h from days 1 to 21 of each 21-day cycle, in conjunction with intravenous administration of pembrolizumab on day 1 of each 21-day cycle. Another arm evaluates patients with locally advanced HER2-negative, hormone receptor-positive breast cancer, adding anastrozole to the above combination. The trial is still ongoing, and there are no results available at this time. The estimated completion date for the trial is September 2023 [236].

#### 6.3.2. Anti-PD-L1 with Cyclin-Dependent Kinase 4/6 (CDK4/6) Inhibitors


**
*NCT04355858*
**


This trial described above evaluates a group of patients receiving SHR1701 (PD-L1/TGF-βRII inhibitor) and SHR6390 (CDK4/6 inhibitor) in Arm-5D. Currently, in phase 2, the first results are expected in May 2023 [235].


**
*NCT02791334*
**


Phase 1a/1b clinical investigation of a new anti-PD-L1 checkpoint antibody, LY3300054, administered either as a standalone therapy or in conjunction with other agents in patients with resistant, advanced solid tumors (Phase 1a/1b anti-PD-L1 combinations in PACT tumors). One arm of the trial evaluates the combination of LY3300054 and abemaciclib in patients with hormone receptor-positive, HER2-negative breast cancer. LY3300054 is administered intravenously on day 1 and day 15, while abemaciclib is administered orally every 12 h throughout a 28-day cycle. The estimated completion date for the trial is December 2023 [237].


**
*NCT03280563*
**


A Phase Ib/II, Open-Label, Multicenter, Randomized Umbrella Study Assessing the Efficacy and Safety of Multiple Immunotherapy-Based Treatment Combinations in Patients with Hormone Receptor-Positive HER2-Negative Inoperable Locally Advanced or Metastatic Breast Cancer (MORPHEUS-HR+ Breast Cancer). This trial is investigating the combination of an anti-PD-L1 antibody (atezolizumab) with a CDK4/6 kinase inhibitor (abemaciclib) and fulvestrant (chemotherapy). Atezolizumab will be given as 840 mg via intravenous (IV) infusion on days 1 and 15 of each 28-day cycle. Atezolizumab will be given as 1200 mg via IV infusion on day 1 of each 21-day cycle. Fulvestrant will be given as 500 mg intramuscularly on days 1 and 15 of Cycle 1 and thereafter on day 1 of each 28-day cycle. Abemaciclib will be given as 150 mg twice daily during each 28-day cycle. The trial is still ongoing, and there are no results available at this time. The estimated completion date for the trial is December 2023 [238].


**
*NCT03147287*
**


In this research study, the investigators are assessing the efficacy of different treatment combinations in participants with HER2-negative hormone receptor-positive metastatic breast cancer who have previously shown resistance to prior palbociclib and endocrine therapy. The treatment regimens being evaluated include fulvestrant alone, fulvestrant in combination with palbociclib, or fulvestrant, palbociclib, and avelumab combined. Palbociclib is administered orally once daily for 21 days within a 28-day cycle. Fulvestrant is administered through two intramuscular injections on cycle 1, days 1 and 15, followed by monthly injections thereafter. Avelumab is administered intravenously once every 2 weeks. The trial is still ongoing, currently in phase 2, and there are no results available at this time. The estimated completion date for the trial is December 2024 [239].


**
*NCT04841148*
**


This trial evaluates the combination of an anti-PD-L1 antibody (avelumab) with a CDK4/6 kinase inhibitor (palbociclib) and hydroxychloroquine (HCQ) in patients with HR+ early-stage BC. Patients will receive palbociclib at a daily oral dose of 125 mg on days 1 to 21, in combination with avelumab administered intravenously at a dose of 10 mg/kg on days 1 and 15 of each 28-day cycle. Alternatively, they may receive a daily oral dose of a 75 mg capsule of palbociclib on days 1 to 28, concurrently with HCQ. Avelumab will be administered at a dose of 10 mg/kg intravenously on days 1 and 15 of each 28-day cycle. The trial is still ongoing, currently in phase 2, and there are no results available at this time. The estimated completion date for the trial is May 2028 [240].


**
*NCT04360941*
**


This trial evaluates the combination of an anti-PD-L1 antibody (avelumab) with a CDK4/6 kinase inhibitor (palbociclib) in patients with HR+/HER2− advanced BC and TNBC. Currently, in phase 1, the first results are expected in June 2024 [241].


**
*NCT03573648*
**


This clinical trial is assessing the combination of an anti-PD-L1 antibody, avelumab, with a CDK4/6 kinase inhibitor, palbociclib, and endocrine therapy in patients with estrogen receptor-positive breast cancer. Eligible patients will undergo a biopsy and be randomly assigned in a 1:2 ratio to receive either endocrine therapy (ET) alone or endocrine therapy with palbociclib (PET). Following the first treatment cycle (28 days), another biopsy will be conducted, and both groups will receive avelumab (A) for an additional three cycles. Subsequently, patients will undergo breast surgery. Throughout the treatment, patients will receive endocrine therapy with or without palbociclib (125 mg orally daily for 21 days followed by 7 days off) for one cycle, with a repeat biopsy, MRI, and blood draw performed on cycle 2, day 1. Avelumab will be subsequently incorporated into both therapeutic groups. Avelumab will be delivered intravenously at a dosage of 10 mg/kg every 14 days, comprising two doses within one cycle, totaling 28 days. Patients will receive a total of three cycles of avelumab in conjunction with endocrine therapy, with or without palbociclib, resulting in a cumulative of four cycles, inclusive of the initial run-in period without avelumab. The specific form of endocrine therapy administered will be determined based on the patient’s menopausal status and will follow established treatment protocols. Premenopausal women will receive daily oral tamoxifen (20 mg for 28 days) in addition to either intramuscular leuprolide (3.75 mg) or subcutaneous goserelin (3.6 mg) on the first day of each cycle. Postmenopausal women will be prescribed daily oral letrozole (2.5 mg). Treatment will persist as long as there is no clinical indication of disease advancement and the treatment is well-received. Individuals who manifest a 25% escalation in tumor size suggestive of disease progression will cease their participation in the study, and they will proceed to undergo final assessments, which encompass a repeat MRI and a surgical intervention. Conversely, individuals who successfully complete all four therapy cycles will undergo MRI scans and the scheduled surgical procedure [242].

All of the above is summarized in Table 5 and Figure 4.

## 7. Discussion

Each day, the pursuit of an efficacious treatment for cancer intensifies. Inhibitors targeting kinases with aberrant activity in cancer stand as a promising approach to combat this affliction. Nonetheless, the intricate network of interplays and the kinetics of kinases often thwart the complete efficacy of most studies. The employment of single-molecule kinase inhibitors frequently leads to the acquisition of resistance to such therapies, alongside a substantial degree of cytotoxicity and numerous adverse effects.

These proteins play a pivotal role in the regulation of cellular signaling and, as a result, are significant therapeutic targets in cancer. Inhibiting these kinases can have beneficial effects by restraining the growth and survival of tumor cells. However, this inhibition, particularly of kinases governing multiple biological processes, can give rise to substantial unwanted side effects.

Taking this fact into consideration and utilizing tools such as network biology studies, GO (Gene Ontology), and KEGG (Kyoto Encyclopedia of Genes and Genomes) analyses, we can comprehend how signaling pathways and biological processes may be interconnected and how the inhibition of certain kinases can trigger cascading effects across multiple cellular pathways.

Specifically, EGFR inhibitors block the activity of this membrane receptor, which plays a pivotal role in the regulation of cell growth and proliferation. This can decelerate cell division and reduce angiogenesis in tumors while simultaneously interfering with the metabolism of healthy cells. EGFR signaling is involved in the regulation of glucose uptake in cells, so its inhibition can impact glucose metabolism across various processes and metabolic pathways. In fact, such inhibition can alter the translocation of GLUT4 to the plasma membrane, thereby reducing glucose uptake by cells. It can also affect the regulation of key enzymes involved in the glucose metabolic pathway, especially the activity of hexokinase, the enzyme catalyzing the initial step of glycolysis, potentially leading to a decrease in glucose uptake and phosphorylation within the cell. Furthermore, this inhibition can also interfere with insulin receptor signaling, which is crucial for regulating glucose uptake by cells, resulting in reduced glucose uptake, decreased insulin sensitivity, and consequently elevated blood glucose levels. This condition of hyperglycemia can also be a consequence of the disruption of key gluconeogenesis enzymes, such as pyruvate carboxylase, phosphoenolpyruvate carboxykinase, and glucose-6-phosphatase, as a result of the decreased EGFR activity. Finally, EGFR is also implicated in the activation of the PI3K/Akt pathway, which is pivotal for the regulation of glucose uptake and cell survival, so its inhibition has an impact.

In summary, the inhibition of EGFR can impact glucose metabolism at multiple levels, including glucose uptake, insulin signaling, the activation of specific signaling pathways, and the regulation of gluconeogenesis. This can lead to an imbalance in glycemic homeostasis and potentially contribute to hyperglycemia in certain conditions [243,244]. However, it is important to note that specific effects may vary depending on the context, such as cell type and the individual’s health status. Patients receiving these medications often require regular monitoring of blood glucose levels and, in some cases, may need to adjust their diabetes treatment if hyperglycemia becomes significant.

Another group of inhibitors that can impact metabolism in various key ways are those targeting the mTOR kinase protein. This inhibition reduces the activation of mTORC1 and mTORC2, leading to a decrease in protein synthesis and cellular growth inhibition. This inhibition affects the PI3K/Akt/mTOR signaling pathway, which plays a crucial role in cellular metabolism regulation. This can result in a reduction in glucose uptake and an imbalance in glycemic homeostasis, as previously described. Regarding lipid metabolism regulation, mTOR inhibition can, on the one hand, increase lipolysis, as it leads to the heightened activation of AMP-activated protein kinase (AMPK), which subsequently activates hormone-sensitive lipase (HSL) and adipose triglyceride lipase (ATGL). These enzymes break down triglycerides into fatty acids, which can then be released into the bloodstream and utilized as an energy source. On the other hand, inhibiting mTOR reduces lipid synthesis, as mTOR positively regulates lipid synthesis, especially the synthesis of fatty acids. When mTOR is active, it promotes the activity of enzymes such as acetyl-CoA carboxylase (ACC) and fatty acid synthase (FAS), both involved in the conversion of acetyl-CoA into fatty acids. The effects of inhibition can impact systemic lipid homeostasis as mTOR regulates the synthesis of very low-density lipoproteins (VLDL) in the liver, carriers of triglycerides and cholesterol through the bloodstream. Therefore, inhibiting mTOR in the liver can decrease VLDL production and, consequently, reduce circulating triglyceride levels in the blood.

Hence, it is essential to consider all of this, as mTOR inhibition can enhance lipolysis and reduce lipid synthesis, potentially resulting in elevated levels of circulating fatty acids in the blood. This, in turn, can have effects on lipid homeostasis and energy metabolism within the body, as well as increase the risk of cardiovascular diseases [245].

Therefore, the need to incorporate new biological treatments with fewer toxic effects is crucial. Immunotherapy, the fifth treatment modality against cancer, complements surgery, radiotherapy, chemotherapy, and molecularly targeted therapies and plays a fundamental role in this regard.

Monoclonal antibodies are an important avenue for improving current therapies in immunotherapy. However, the use of monoclonal antibodies in monotherapy is not as effective as expected, despite the biological advantage of using these antibodies due to the few adverse effects. Therefore, combining monoclonal antibodies with kinase inhibitors is a fertile area of research in breast cancer. Bispecific antibodies (BsAB) also show promise in treating unmet needs in breast cancer, such as de-escalation of chemotherapy, treatment of HR+ disease, and treatment of residual disease after neoadjuvant therapy. New clinical and preclinical BsAB are under development, with most of them focused on HER2+ and scarce in other breast cancer subtypes. Combined treatments of autologous T cells, chemotherapy, BsAB-based ADCs, nanotechnologies, and other immunotherapeutics may be promising, with the aim of establishing a personalized therapy.

Enhanced comprehension of cancer cells’ immune evasion mechanisms and the identification of specific immune checkpoint inhibitors have ushered in novel prospects for therapeutic interventions. However, monotherapy with monoclonal antibodies against PD-1 and PD-L1 has shown limited effectiveness in the treatment of metastatic breast cancer, partly due to the low number of tumor-infiltrating lymphocytes in most breast cancers and the adverse effects they manifest in the metabolic pathways of immune cells.

Thus, by inhibiting immune checkpoint proteins, T cells become more active. Like other cell types, they obtain energy through glycolysis, a metabolic process involving the breakdown of glucose to produce ATP. Glucose is transported into T cells through glucose transporters such as GLUT1. This leads to an increase in glycolysis in activated T cells and an upregulation of GLUT1 expression. This, in turn, provides the necessary energy to perform effective functions, such as cytokine production and the lysis of cancer cells. This increased glycolysis can result in higher lactate production, which can impact the acidification of the tumor microenvironment. In general, cells, by enhancing glycolysis, take up more glucose from the bloodstream to meet their energy requirements. This can lead to short-term reductions in blood glucose levels. However, there is a hormonal counter-regulation; when T cells and other types of immune cells are activated due to immunotherapy with immune checkpoint inhibitors, they can release cytokines and other molecules that trigger a counterregulatory hormonal response. This hormonal response, which includes the release of hormones such as glucagon and cortisol, can increase glucose production in the liver and decrease glucose uptake by peripheral tissues. Additionally, in some cases, immune activation induced by immunotherapy can lead to insulin resistance. Insulin resistance can cause glucose to remain in the blood rather than enter cells, contributing to hyperglycemia [246].

The use of combination therapies could be a breakthrough in treatment as it would take into account the specificities of each therapy separately; kinase inhibitors are small enough to interfere with intracellular signaling pathways in cancer cells, while antibodies generally cannot penetrate a cell but do affect intercellular signaling pathways, targeting key cells of the immune system. Several studies suggest that combination therapies with kinase inhibitors, immune checkpoint inhibitors, and monoclonal antibodies have a synergistic effect that achieves better response rates and could, therefore, be implemented in early stages as well as in advanced and metastatic stages, changing the treatment paradigm.

The trials currently underway and those expected to yield results in the coming years are evaluating combinations of promising therapies. As shown in Table 4, these trials combine checkpoint inhibitors (anti-PD-1/PD-L1), kinase inhibitors, chemotherapeutic drugs, and endocrine therapy as they are targeted at patients with HR+ breast cancer. While the specific effects may vary depending on the patient and clinical context, we can hypothesize about the expected effects of each combination. The rationale, based on the mechanism of action of each therapy, is to enhance the immune response and inhibit cell proliferation.

There are notable differences in the case of kinase inhibitors. Abemaciclib, in addition to inhibiting CDK4/6, also inhibits CDK9 kinase. This has an additional effect on cell cycle regulation and the suppression of gene transcription involved in cell proliferation. Therefore, abemaciclib may have a broader spectrum of activity compared to palbociclib. Palbociclib specifically blocks CDK4/6 kinases, but its primary focus is to inhibit the retinoblastoma (Rb) pathway signaling, which is involved in cell cycle control. By inhibiting CDK4/6, palbociclib blocks Rb phosphorylation and prevents its release, resulting in cell growth inhibition.

If we consider toxicity criteria and analyze it theoretically, the combination of atezolizumab, abemaciclib, and fulvestrant (study NCT03280563) could potentially have lower cytotoxicity compared to other combinations that include chemotherapy. This is because atezolizumab and abemaciclib are immunotherapeutic and kinase inhibitors, respectively, and do not belong to the traditional category of cytotoxic chemotherapy. On the other hand, combinations that include chemotherapy, such as paclitaxel and aromatase inhibitors or anastrozole, could potentially have a broader toxicity profile due to the general cytotoxic effects of chemotherapy. Similarly, combinations that include multiple kinase inhibitors, such as SHR6390, abemaciclib, and palbociclib, could increase the risk of toxicity due to the potential overlap of kinase inhibitory effects and associated side effects.

It is important to note that toxicity is an aspect that should be carefully evaluated and monitored by the medical team. The magnitude and exact nature of the side effects or toxicity of a specific therapeutic combination can vary in each patient and depend on various factors, such as the administered dose, treatment duration, and individual patient response.

Based on clinical practice, it could be predicted that, as mentioned earlier, the group of HR+/HER2− patients in both local, advanced, or metastatic cancer would be the most suitable for receiving the combination therapies detailed in Table 4. Therefore, the combination of atezolizumab, abemaciclib, and fulvestrant (study NCT03280563) could be considered the clinical therapy with the most benefits and potentially less toxicity, as it would achieve the primary goal of disease control and preservation of quality of life. Another factor to consider is the application of therapy to patients with better overall health and organ function, as these patients may have better tolerance to treatments and lower susceptibility to potential side effects.

In the research and development of therapies, striking a balance between therapeutic efficacy and minimizing side effects is essential. This may involve designing more selective combination therapies or devising strategies to mitigate collateral effects. It is also imperative to conduct rigorous clinical trials to assess the safety and effectiveness of these therapy combinations in patients. In summary, the synergy between kinase inhibition and immunotherapy is an active area of research in the field of cancer treatment. While promising, the combination of these therapies presents challenges and limitations that require careful consideration and development.

## 8. Conclusions

In summary, the quest for determining the most effective combination for individual patients necessitates the validation of predictive biomarkers within the tumor and its adjacent microenvironment, a comprehensive grasp of the pharmacokinetics and pharmacodynamics of these drug pairings, and the refinement of dosages to enhance clinical outcomes. This applies not only to patients with advanced-stage cancers but also to those in earlier disease stages who face a heightened risk of disease recurrence.

## Figures and Tables

**Figure 1 cancers-15-05499-f001:**
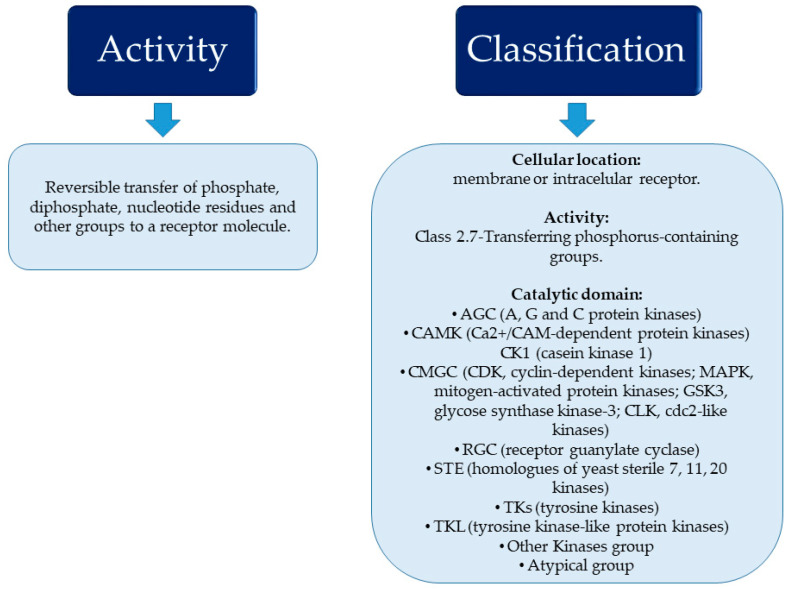
General information about protein kinases. Description of their activity and classification according to cellular location, activity, and catalytic domains. Adapted with permission from Reference [24]. Copyright 2019, copyright García-Aranda M, Redondo M.

**Figure 2 cancers-15-05499-f002:**
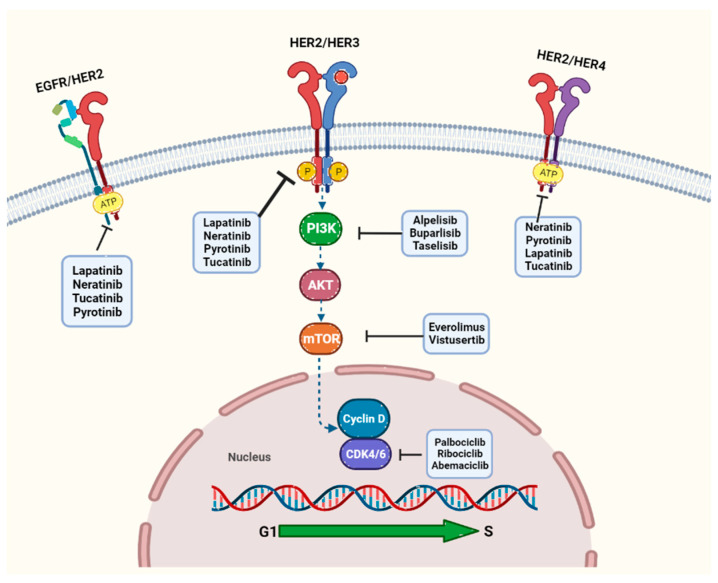
Tyrosine kinase inhibitors and implicated receptors. The mechanism of action of combination therapy of kinase inhibitors with other drugs, as well as the receptors involved in different trials, is detailed here. Images were created using Biorender.com (accessed on 14 September 2023).

**Figure 3 cancers-15-05499-f003:**
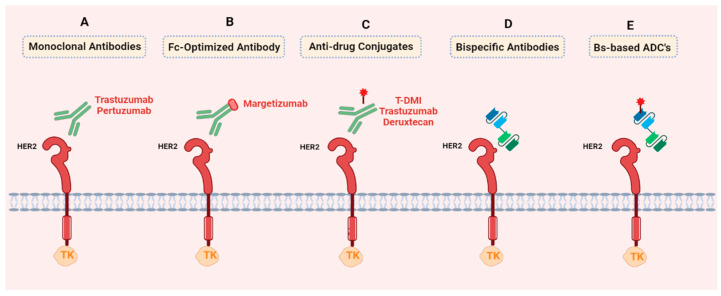
Mechanism of action of the different HER2 antibodies. (**A**,**B**) Monoclonal antibodies that are directed against epitopes present in the external domain of HER2 receptors. (**C**) Antibody–drug conjugates (ADCs): antibody directed to a target antigen; the payload, a cytotoxic agent; and a linker, which connects the antibody to the payload. (**D**) Bispecific antibodies. (**E**) A combination of ADCs and bispecific antibodies. The image was created using Biorender.com (accessed on 14 September 2023).

**Figure 4 cancers-15-05499-f004:**
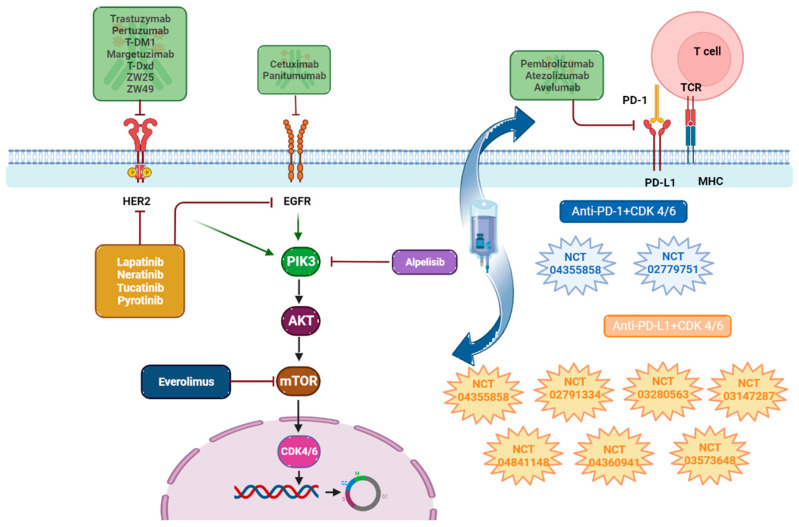
Anti-PD-1/PD-L1 with cyclin-dependent kinase 4/6 (CDK4/6) inhibitors. Various trials are evaluating the combination of antibodies against PD-1/PD-L1 with CDK4/6 kinase inhibitors, along with other chemotherapy drugs. Image was created using Biorender.com. https://app.biorender.com/illustrations/6408b8a6a9e401e58f089c31 (accessed on 28 March 2023).

**Table 1 cancers-15-05499-t001:** Molecular classification of breast cancer.

Subtypes		
HR+	This subtype accounts for up to 75% of breast cancer tumor cases [9]	**Luminal-A** Represent 50–60% of all breast cancers. They are defined as ER+ and/or PR+, HER2−, and low Ki67 (<14%) [8,9]. **Luminal-B** Represent 15–20% of breast cancers. They are defined as ER+ and/or PR+/− (PR < 20% + Ki67 ≥ 14%) with HER2− as well as ER+ and/or PR+/− (any PR+ and any Ki67) and HER2+ [8,10].
HER2-Enriched	These tumors are defined as ER−, PR−, and HER2+ [10].	
Basal-Like	TNBC tumors, which constitute approximately 80% of the basal-like tumors [11] and account for 15–20% of breast carcinomas [12]. They are defined as ER−, PR−, HER2−, CK5/6+, and/or EGFR+ [10].	

HR+: hormone receptors positive. ER+: estrogen receptors positive. PR+: progesterone receptors positive. HER2+: positive for human epidermal growth factor receptor 2 (receptor tyrosine-protein kinase ERBB2, CD340). CK5/6+: expressing cytokeratin 5/6. EGFR+: expressing epidermal growth factor receptor.

**Table 2 cancers-15-05499-t002:** Protein kinases and subtype breast cancer.

Altered Protein Kinase	Subtype of Breast Cancer
HER2	HER2-enriched
PI3K/Akt/mTOR	HER2-enriched Hormone receptor-positive
PI3K	HER2-enriched Hormone receptor-positive
mTOR	HER2-enriched Estrogen receptor-positive HER2-negative Hormone receptor-positive
PTEN	HER2-enriched Basal-like
PDK	HER2-enriched
Akt	Estrogen receptor-negative HER2-enriched Hormone receptor-positive
PDPK1	Hormone receptor-positive
MAPK	Hormone receptor-positive Basal-like
AURK	Hormone receptor-positive Basal-like
CDK	Hormone receptor-positive
PLK1	Hormone receptor-positive
SK	Basal-like
PTK	HER2-enriched

**Table 3 cancers-15-05499-t003:** Summary of current specific/selective tyrosine kinase inhibitors used in breast cancer therapy.

Inhibitor Kinase	Combination Therapy	Subtype of Breast Cancer	Trial Name or Registration Number	Adverse Events
HER2, HER1 Receptor (EGFR) Inhibitors				
Lapatinib	Capecitabine	HER2+, pre-treated with anthracycline, taxane and trastuzumab	NCT01050322 GLICO-0801	Diarrhea, hand-foot syndrome, nausea, rash, and fatigue
	Letrozole	Menopausal with HR+ and HER2+, metBC	NCT00073528	
Neratinib	Capecitabine	HER1, HER2, HER4. Adjuvant early-state HER2+ pre-treated trastuzumab BC	NCT00878709 ExteNET	Diarrhea, vomiting, nausea
Tucatinib	Capecitabine plus trastuzumab	HER2+, unresectable mBC	NCT02614794 HER2CLIMB	Diarrhea, palmar-plantar erythrodysesthesia, nausea, fatigue, and rash
Pyrotinib (Non FDA)	Capecitabine after previous trastuzumab	HER1, HER2, HER4, mBC	NCT03080805 PHOEBE	Diarrhea, palmar-plantar erythrodysesthesia
Cyclin-dependent kinase 4/6 inhibitors				
Palbociclib	Letrozole	HR+/HER2, postmenopausal mBC	NCT01740427 PALOMA-2	Neutropenia, fatigue, anemia, nausea, alopecia
	Fulvestrant	HR+/HER2−, mBC	NCT01942135 PALOMA-3	
Ribociclib	Letrozole	HR+, HER2−, postmenopausal mBC	NCT01958021 MONALEESA-2	Neutropenia, nausea and infection
	Fulvestrant	HR+, HER2−, premenopausal advanced mBC	NCT02422615 MONALEESA-3 NCT02278120 MONALEESA-7	Leukopenia, tiredness, and nausea
Abemaciclib	Letrozole	HR+, HER2−, mBC	NCT02246621 MONARCH 3	Diarrhea
PI3K inhibitors				
Buparlisib	Fulvestrant Paclitaxel	HR+/HER2−, mBC	NCT01610284 BELLE-2 NCT01572727 BELLE-4	Neutropenia
Alpelisib	Fulvestrant	HR+/HER2−, advanced BC	NCT02437318 SOLAR-1	Hyperglycemia, severe skin reaction, and diarrhea
Taselisib	Letrozole	HR+/HER2−, early BC	NCT02273973 LORELEI	Diarrhea, nausea and fatigue
mTOR inhibitors				
Everolimus	Fulvestrant	HR+, HER2−, mBC	NCT02216786 MANTA	Stomatitis, anemia, dyspnea, hyperglycemia, fatigue, and pneumonitis
	Trastuzumab	Trastuzumab-resistant and taxane pre-treated, HER2+, mBC	NCT01007942 BOLERO-3	
Vistusertib (AZD2014)	Palbociclib plus Fulvestrant	HR+, mBC	NCT02599714 PASTOR	---

Hormone receptor-positive: HR+; HER2-negative: HER2−; HER2-positive: HER2+; BC: breast cancer; mBC: metastatic breast cancer.

**Table 4 cancers-15-05499-t004:** Current clinical trials of ADCs and bispecific antibodies in HER2+ breast cancer. https://clinicaltrials.gov (accessed on 14 September 2023).

ADC	Payload	Subtype of Breast Cancer	Trial Registration Number
ARX788	Amberstatin269 (AS269)	HER2+ BC	NCT01042379
		HER2+ mBC	NCT04829604 NCT02512237
		HER2-mutated	NCT05041972
		HER2-low BC	NCT05018676
		Brain mBC	NCT05018702
		HER2+ solid tumor	NCT03255070
Disitamab vedotin (RC48)	MMAE	HER2+ advanced, mBC	NCT02881190
		HER2+ BC	NCT05134519
		Locally advanced or metastatic HER2-low BC	NCT04400695
		HER2-expression mBC with abnormal activation of the PAM pathway	NCT05331326
		Advanced BC	NCT03052634
		HER2-low BC	NCT05726175
		HER2+ mBC with or without liver metastases	NCT03500380
A166	Duo-5	HER2+ pretreated BC	NCT03602079
MRG002	MMAE	HER2+ advanced BC	NCT05263869
		HER2+ metastatic tumors	NCT04924699
		HER2-low locally advanced mBC	NCT04742153
BDC-1001	TLR7/8 agonist	HER2+ advanced BC	NCT04278144
ALT-P7	MMAE	HER2+ BC	NCT03281824
XMT-1522	AF-HPA	Advanced HER2+ BC patients	NCT02952729
PF-06804103	Derivative of auristatin	HER2+ BC	NCT03284723
TTCs/BAY2701439	Thorium-227	Advanced HER2-expressing cancer	NCT04147819
Bispecific Ab	In combination with	Subtype of breast cancer	Trial registration number
Zanidatamab (ZW25)	Chemotherapy	Pre-treated advanced HER2+ BC	NCT02892123
Zanidatamab (ZW25) MBS301		HER2+ Early BC	NCT05035836
	Palbociclib and fulvestrant	HER2+ advanced BC	NCT04224272
	Anti-CD47	HER2+ solid tumors/HER2-low BC	NCT05027139
		HER2+ solid tumors	NCT03842085
KN026	Chemotherapy	HER2+ BC	NCT04881929
KN026 Zenocutuzumab (MCLA-128)	* KN046	Locally advanced HER2+ solid tumors and HER2+ solid tumor	NCT04521179 NCT04040699
	Palbociclib and fulvestrant	Advanced BC	NCT04778982
	Trastuzumab + chemotherapy or trastuzumab and vinorelbine	HER2-low BC HER2+ mBC	NCT03321981
MM-111	Trastuzumab	Advanced HER2 amplified and heregulin-positive BC	NCT01097460
MM-111 Ertumaxomab		Advanced, refractory HER2 A\amplified and heregulin-positive cancers	NCT00911898
		HER2-high or HER2-low BC	

* KN046: bispecific antibody against PD-1 and CTLA-4; MMAE: monomethyl auristatin E; AF-HPA: auristatin derivative; DUO-5: duostatin-5.

**Table 5 cancers-15-05499-t005:** Immunotherapy against PD-1/PD-L1 and kinase inhibitor treatment in breast cancer.

Target	Checkpoint Inhibitors	Kinase Inhibitor	Combined Therapy	Subtype of Breast Cancer	Trial Number	Phase
PD-1	SHR1210 (PD-1 antibody)	SHR6390 (CDK4/6 inhibitor)	Paclitaxel, fulvestrant, aromatase inhibitor (chemotherapy)	mBC	NCT04355858	2
	Pembrolizumab	Abemaciclib	Anastrozole (chemotherapy)	HR+/HER2− Locally Advanced or mBC	NCT02779751	1
PD-L1	SHR1701 anti-PD-L1/TGF-βRII bifunctional fusion protein	SHR6390 (CDK4/6 inhibitor)	Paclitaxel, fulvestrant, aromatase inhibitor (chemotherapy)	HR+/HER2− advanced and mBC	NCT04355858	2
	Anti-PD-L1 Checkpoint Antibody (LY3300054)	Abemaciclib	Chemotherapy	HR+/HER2− BC	NCT02791334	1a/1b
	Atezolizumab	Abemaciclib	Fulvestrant	HR+/HER2− Inoperable locally advanced or mBC	NCT03280563	1
	Avelumab	Palbociclib	Fulvestrant	HR+/HER2− Endocrine Pre-treated mBC	NCT03147287	2
			HCQ (hydroxychloroquine)	HR+, early-stage BC	NCT04841148	2
			-	HR+/HER2− advance BC, TNBC	NCT04360941	1
			Endocrine therapy	HR + BC	NCT03573648	2

Hormone receptor-positive = HR+; breast cancer = BC; HER2-negative = HER2−; metastatic breast cancer = mBC; triple-negative breast cancer = TNBC.

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
