# Peer review of "Unlocking New Avenues in Breast Cancer Treatment: The Synergy of Kinase Inhibitors and Immunotherapy"

_cancers, 2023, doi:10.3390/cancers15235499_

Round 1

Reviewer 1 Report

Comments and Suggestions for Authors

The manuscript by Bravo Romero et al. deals with a topic that is widely studied and discussed in the literature. The review is comprehensive and even too much detailed when discussing classical topics like protein kinase families.

I think the manuscript could be of interest to a wide audience of readers and scientists and deserves publication, even though some aspect should be slightly revised.

The description of each protein kinase subfamily is widely acknowledged and can be found even in text books, for this reason I suggest to shorten that section and to include a more succinct description of the kinase activities while discussing the relative inhibitors.

However, the manuscript is overall well written and easily readable and, with the suggested revisions, I think it can be published in Cancers.

Author Response

The manuscript by Bravo Romero et al. deals with a topic that is widely studied and discussed in the literature. The review is comprehensive and even too much detailed when discussing classical topics like protein kinase families.

I think the manuscript could be of interest to a wide audience of readers and scientists and deserves publication, even though some aspect should be slightly revised.

The description of each protein kinase subfamily is widely acknowledged and can be found even in text books, for this reason I suggest to shorten that section and to include a more succinct description of the kinase activities while discussing the relative inhibitors.

However, the manuscript is overall well written and easily readable and, with the suggested revisions, I think it can be published in Cancers.

We wholeheartedly agree with your suggestions and are appreciative of them. While we have summarised the descriptions of the various subfamilies of protein kinases, as you have indicated, we also believe that a concise description is necessary to facilitate a better understanding of the mechanisms of action of the inhibitors of these proteins and their synergy with immune checkpoint inhibitors, as detailed in crucial sections of this review.

Reviewer 2 Report

Comments and Suggestions for Authors

The synergy between kinase inhibitors and immunotherapy is a promising area of research in cancer treatment. This synergy arises from their complementary mechanisms of action. Kinase inhibitors can help prime the tumor microenvironment and make it more accessible to the immune system. They can reduce the immunosuppressive signals produced by cancer cells, such as inhibitory cytokines. This makes it easier for immunotherapy to work effectively.

The authors described the characteristics of the main protein kinase targets in breast cancer, such as EGFR, AKT, mTOR, and so on. The authors also listed and described the characteristics of specific inhibitors of these proteins, both organic molecules, such as chemical compounds, and antibodies.

From my standpoint, the review emerges as a meticulously organized, extensively detailed, and adeptly presented inventory of the various actors involved in the process of kinase inhibition, encompassing the immunotherapeutic aspects as well. It is a precise description, but it describes the macroscopically measurable effects of the problem.

What I mean. The reader understands very well that using an inhibitor cancels the action of the target protein, but the question certainly arises: where does all this happen?

Researchers widely study these proteins (such as EGFR or mTOR) in network biology, which today represents one of the few, if not the only, approaches to delve into the world of metabolism at the molecular level. These proteins represent important nodes of metabolic processes. Without wanting to go into the details of the networks, the metabolic world in which these inhibitory actions occur is the deep metabolic world. What we call mesoscopic, where the vision of what is happening metabolically at this level, is a little different from the static one we usually used to discuss cancer. These molecules are part of processes that have very specific spatio-temporal characteristics. These proteins are nodes that manage and control many functional relationships with other nodes (hubs). Sometimes they do not have many functional relationships but they control important functional crossroads on different metabolic paths, acting as traffic lights on intersections (Bottleneck nodes). In short, these particular nodes control not 1 or 2 or 3 biological processes but hundreds of each. A simple Biological Processes (GO) analysis or KEGG analysis clearly shows what I am saying. This opens a discussion of the adverse effects of these inhibitors. For example, EGFR (Epidermal Growth Factor Receptor) is involved in dozens of different metabolic events that all occur at different metabolic times. It is a dynamic system, not a static one. This can lead to "collateral" errors at the molecular level, which we macroscopically show as:

Metabolic Reprogramming: Cancer cells often undergo metabolic reprogramming to support their rapid growth and survival. This can include changes in glucose metabolism (the Warburg effect) and alterations in amino acid metabolism. Kinase inhibitors may affect these metabolic pathways by disrupting signaling pathways useful for cells. Thus, some kinase inhibitors may have adverse metabolic effects. We observe these effects at a macroscopic level. For example, they can impact glucose metabolism, potentially leading to hyperglycaemia or exacerbating existing metabolic conditions as diabetes. Some kinase inhibitors can influence lipid metabolism or other metabolic pathways. But we must explain their profound origin, as far as possible.

Or even,

Immunometabolism: Immunotherapy, particularly checkpoint inhibitors, can also influence metabolic pathways in immune cells. Immune cells require specific metabolic adaptations to function effectively. For example, activated T cells need to shift their metabolism from oxidative phosphorylation to glycolysis to support their effector functions. Thus, the combination of kinase inhibitors and immunotherapy can lead to synergistic effects. For instance, some kinase inhibitors may reduce the immunosuppressive signals produced by cancer cells, potentially creating a more favourable metabolic environment for immune cells to attack the tumor. But, the metabolic effects of these treatments can vary between patients because of individual differences in metabolism, the specific cancer type, and the chosen treatment regimen. Authors should address these aspects, even if to a limited extent.

I encourage the authors to elucidate the profound aspects of macroscopic observations to prevent the outcome from being a monotonous compilation of names and facts, which will be hard to retain and communicate as a unified perspective.

When composing a review, it is essential for the author to situate their assertions within the realm of collective knowledge concerning the sector, while also acknowledging its constraints and interdependencies. Moreover, expressing one's opinion and offering plausible predictions are key components of a thorough review. Otherwise, what's the point of a review? It's easy to make a list of events, it's more difficult to comment on it.

Author Response

The synergy between kinase inhibitors and immunotherapy is a promising area of research in cancer treatment. This synergy arises from their complementary mechanisms of action. Kinase inhibitors can help prime the tumor microenvironment and make it more accessible to the immune system. They can reduce the immunosuppressive signals produced by cancer cells, such as inhibitory cytokines. This makes it easier for immunotherapy to work effectively.

The authors described the characteristics of the main protein kinase targets in breast cancer, such as EGFR, AKT, mTOR, and so on. The authors also listed and described the characteristics of specific inhibitors of these proteins, both organic molecules, such as chemical compounds, and antibodies.

From my standpoint, the review emerges as a meticulously organized, extensively detailed, and adeptly presented inventory of the various actors involved in the process of kinase inhibition, encompassing the immunotherapeutic aspects as well. It is a precise description, but it describes the macroscopically measurable effects of the problem.

What I mean. The reader understands very well that using an inhibitor cancels the action of the target protein, but the question certainly arises: where does all this happen?

Researchers widely study these proteins (such as EGFR or mTOR) in network biology, which today represents one of the few, if not the only, approaches to delve into the world of metabolism at the molecular level. These proteins represent important nodes of metabolic processes. Without wanting to go into the details of the networks, the metabolic world in which these inhibitory actions occur is the deep metabolic world. What we call mesoscopic, where the vision of what is happening metabolically at this level, is a little different from the static one we usually used to discuss cancer. These molecules are part of processes that have very specific spatio-temporal characteristics. These proteins are nodes that manage and control many functional relationships with other nodes (hubs). Sometimes they do not have many functional relationships but they control important functional crossroads on different metabolic paths, acting as traffic lights on intersections (Bottleneck nodes). In short, these particular nodes control not 1 or 2 or 3 biological processes but hundreds of each. A simple Biological Processes (GO) analysis or KEGG analysis clearly shows what I am saying. This opens a discussion of the adverse effects of these inhibitors. For example, EGFR (Epidermal Growth Factor Receptor) is involved in dozens of different metabolic events that all occur at different metabolic times. It is a dynamic system, not a static one. This can lead to "collateral" errors at the molecular level, which we macroscopically show as:

Metabolic Reprogramming: Cancer cells often undergo metabolic reprogramming to support their rapid growth and survival. This can include changes in glucose metabolism (the Warburg effect) and alterations in amino acid metabolism. Kinase inhibitors may affect these metabolic pathways by disrupting signaling pathways useful for cells. Thus, some kinase inhibitors may have adverse metabolic effects. We observe these effects at a macroscopic level. For example, they can impact glucose metabolism, potentially leading to hyperglycaemia or exacerbating existing metabolic conditions as diabetes. Some kinase inhibitors can influence lipid metabolism or other metabolic pathways. But we must explain their profound origin, as far as possible.

Or even,

Immunometabolism: Immunotherapy, particularly checkpoint inhibitors, can also influence metabolic pathways in immune cells. Immune cells require specific metabolic adaptations to function effectively. For example, activated T cells need to shift their metabolism from oxidative phosphorylation to glycolysis to support their effector functions. Thus, the combination of kinase inhibitors and immunotherapy can lead to synergistic effects. For instance, some kinase inhibitors may reduce the immunosuppressive signals produced by cancer cells, potentially creating a more favourable metabolic environment for immune cells to attack the tumor. But, the metabolic effects of these treatments can vary between patients because of individual differences in metabolism, the specific cancer type, and the chosen treatment regimen. Authors should address these aspects, even if to a limited extent.

I encourage the authors to elucidate the profound aspects of macroscopic observations to prevent the outcome from being a monotonous compilation of names and facts, which will be hard to retain and communicate as a unified perspective.

When composing a review, it is essential for the author to situate their assertions within the realm of collective knowledge concerning the sector, while also acknowledging its constraints and interdependencies. Moreover, expressing one's opinion and offering plausible predictions are key components of a thorough review. Otherwise, what's the point of a review? It's easy to make a list of events, it's more difficult to comment on it.

We are genuinely thankful for the insightful suggestions provided, as they have proven to be both captivating and immensely valuable in enhancing the quality of our ongoing review. Consequently, we have incorporated an examination of the impact of inhibiting specific proteins, including kinases and ICI, at the molecular level on metabolism. Throughout this process, we have diligently taken into account the potential adverse effects stemming from the intricate interplay among various metabolic pathways.

 Our efforts in this regard have been pursued to the fullest extent possible, aligning with an active area of research replete with numerous trials currently awaiting results. The invaluable suggestions have provided us with a fresh perspective, enabling a more objective portrayal of the advantages of treatment synergy, while also shedding light on the inherent limitations and potential side effects of this therapeutic approach. As a result, we are now able to offer our own perspective on this matter, albeit within the confines of our review's scope.

Round 2

Reviewer 2 Report

Comments and Suggestions for Authors

I find that the manuscript is now easier to read and better organized. I think it can be published.